# Chemical Composition, Antioxidant and Antimicrobial Activity of Some Types of Honey from Banat Region, Romania

**DOI:** 10.3390/molecules27134179

**Published:** 2022-06-29

**Authors:** Silvia Pătruică, Ersilia Alexa, Diana Obiștioiu, Ileana Cocan, Isidora Radulov, Adina Berbecea, Roxana Nicoleta Lazăr, Eliza Simiz, Nicoleta Maria Vicar, Anca Hulea, Dragoș Moraru

**Affiliations:** 1Faculty of Bioengineering of Animal Resources, Banat’s University of Agricultural Sciences and Veterinary Medicine “King Michael I of România” from Timișoara, Calea Aradului nr.119, 300645 Timișoara, Romania; silviapatruica@usab-tm.ro (S.P.); roxanalazar@usab-tm.ro (R.N.L.); elizasimiz@usab-tm.ro (E.S.); dragos.moraru@usab-tm.ro (D.M.); 2Faculty of Food Engineering, Banat’s University of Agricultural Sciences and Veterinary Medicine “King Michael I of România” from Timișoara, Calea Aradului no. 119, 300645 Timişoara, Romania; ileanacocan@usab-tm.ro; 3Faculty of Veterinary Medicine, Banat’s University of Agricultural Sciences and Veterinary Medicine “King Michael I of România” from Timișoara, Calea Aradului no. 119, 300645 Timişoara, Romania; anca.hulea@usab-tm.ro; 4Faculty of Agriculture, Banat’s University of Agricultural Sciences and Veterinary Medicine “King Michael I of România” from Timișoara, Calea Aradului no. 119, 300645 Timişoara, Romania; isidora_radulov@usab-tm.ro (I.R.); adina_berbecea@usab-tm.ro (A.B.); nicoleta.vicar@usab-tm.ro (N.M.V.)

**Keywords:** honey, chemical composition, antioxidant and antimicrobial activity

## Abstract

Honey is a natural product with multiple health benefits. The paper presents the chemical characterization and the antioxidant and antimicrobial potential of ten types of honey (knotweed, linden, wild cherry, acacia, honeydew, oilseed rape, sunflower, phacelia, plain polyflora and hill polyflora) from the Banat region, Romania. We studied the water content, dry matter, impurities, acidity and pH of honey. We also determined the content of reducing sugar, minerals and flavonoids and the total phenolic content. All honey samples analysed showed good nutritional characteristics according to the standard codex for honey. From the analysis of the mineral content of the honey samples, we observed a variability in the macro and microminerals, influenced by the botanical origin, ranging between 0.25% (wild cherry honey) and 0.54% (honeydew). The toxic metals’ (Cd and Pb) levels met the standard for almost all samples analysed except for knotweed. The flavonoid content of the samples ranged from 9.29 mg QE/100 g for wild cherry honey to 263.86 mg QE/100 g for linden honey, and for polyphenols between 177.6 mgGAE/100 g for acacia honey and 1159.3 mgGAE/100 g for honeydew. The best antioxidant capacity was registered in the case of linden honey (79.89%) and honeydew (79.20%) and the weakest in acacia (41.88%) and wild cherries (50.4%). All studied honey samples showed antimicrobial activity, depending on the type of honey, concentration and strain analysed. The novelty of this study is given by the complex approach of the study of honey quality, both from the perspective of chemical attributes and the evaluation of the antimicrobial potential on specific strains in correlation with the botanical and geographical origin of the analyzed area.

## 1. Introduction

Honey is a natural product resulting from the processing by bees of flower nectar, sweet secretions present on other parts of the plant or excretions produced by some insects (*Hemiptera*), such as aphids, which live and parasitize parts of plants [1].

Consumed worldwide since ancient times for its nutritional and therapeutic properties, honey contains mainly simple carbohydrates (fructose and glucose), water and other compounds such as enzymes, proteins, amino acids, phenolic compounds, organic acids, minerals and vitamins [2,3,4,5]. The chemical composition of honey, and its taste and color, depend mainly on its botanical origin, bee species, climate and geographical region, but may also be influenced by weather conditions and its processing, packaging and storage [6,7,8]. The therapeutic potential of honey is attributed to its antioxidant and antimicrobial capacity [8,9,10,11,12,13], polyphenols being partly responsible for the antioxidant activity of honey [13]. Flavonoids and polyphenols present in honey act as antioxidants against free radicals, preventing the cell’s ageing. Darker honey has a higher polyphenol content [14,15] and a more intense antioxidant activity [16]. There is usually a correlation between the total content of polyphenols and flavonoids and the antioxidant capacity of honey [17] but not always [18,19] because the antioxidant capacity of each sample is the combined result of other non-phenolic compounds [20].

Mračevic et al., 2020, studied the chemical composition and biological activity of seven honey types from different regions of Serbia. The chemical composition of honey showed a significant variability according to their botanical and geographical origin. The antioxidant and antimicrobial activity varied significantly among the honey samples [5].

Honey is known for its antimicrobial potential, having a broad spectrum against microorganisms, including bacteria [21]. The antimicrobial efficacy of honey is influenced by factors such as osmolarity, H_2_O_2_ content, low pH, polyphenol and flavonoid content [21,22,23,24], and these are correlated with the source of nectar and storage conditions [25,26,27].

Numerous studies have evaluated the antimicrobial activity of honey of different botanical and geographical origins [25,26,27,28]. In a recent study, Kolayli et al., 2020, reported [28] the good antimicrobial activity against *S. aureus* in the case of buckwheat honey (*Fagopyrum eculentum*), heather honey (*Calluna vulgaris*), nettleorurtica honey (*Urtica dioica*), thistle honey (*Silybium marianum*), calltrop honey (*Eryngium campestre*), coriander honey (*Coriandrum sativum*), thyme honey (*Thymus vulgaris*) and honeydew. At the same time, they observed a moderate antimicrobial activity in the case of heather honey (*Calluna vulgaris*) and honeydew against *E. coli*, and heather honey (*Calluna vulgaris*) against *C. albicas*. Research conducted by Grego, E., et al., 2016, in the Italian honey [29], highlighted that the antimicrobial activity of honeydew, polyfloral and chestnut honey against *S. aureus* was similar to manuka honey.

According to the data provided by [30], the honey production obtained in Romania in the last 5 years was between 18,000 and 30,000 tons/year, the main types of honey obtained being sunflower, polyflora, acacia, linden, rapeseed and honeydew. More than half of the honey produced in Romania is exported, mainly to European Union countries.

The main objective of this paper was to compare the chemical composition, and antioxidant and antimicrobial activity of different types of honey from the nectar of: (i) honey trees (linden, acacia, wild cherry); (ii) cultivated agricultural plants (sunflower, oilseed rape, phacelia); (iii) flowers from plain and hill meadows (multifloral), of invasive plants (knotweed), but also honey from the excretions produced by some insects, such as aphids (honeydew).

## 2. Results and Discussion

### 2.1. Chemical Composition

Water is the second constituent of honey and the water content provides information on its degree of maturation [31]. The water content of honey depends on the botanical origin, the degree of honey maturation, the processing technique and the storage conditions [32,33,34]. According to [35], honey humidity should not be higher than 20%. Water content affects some characteristics of honey such as maturation, viscosity and crystallization [36]. The high water content causes the honey to ferment, affecting its quality [34,37].

In our study, the honey samples had a humidity between 14.34% (linden honey) and 16.76% (knotweed). With an average of 15.28% (Figure 1), they were concordant with the results obtained in the literature [38,39]. Small differences between the recorded values can be attributed to the geographical region, the maturation temperature and the humidity during the harvest period [5], given the climate change of recent years [40].

Impurities can get into the honey during the spinning and packing process [41]. According to [35], the impurity content of honey must not exceed 100 mg/100 g of honey. In the present study, the impurity content of the samples analysed ranged from 42 mg/100 g sample (acacia honey) to 65 mg/100 g sample (sunflower honey) with an average of 52 mg/100 g honey (Figure 1).

The total ash content of honey depends on its botanical origin, and pedological and climatic conditions [5,42,43]. There is a correlation between the color of honey and the content in mineral salts; dark-colored honey has a higher content of mineral salts compared to light-colored [33,44,45]. The ash content of honey from flower nectar should not exceed 0.6%, and in the case of honeydew 1.2% [35]. The content in mineral salts can be used as a parameter to assess the nutritional value of honey, but also as an indicator of environmental pollution [3].

In the honey samples analysed by us, the content of mineral substances was between 0.25% (wild cherry honey) and 0.54% (honeydew) with an average of 0.32% (Figure 2), and this corresponds to some types of honey in the data obtained by [5].

The mineral content of floral honey is around 0.1–0.2% and of honeydew around 1% and depends on the botanical origin, and pedological and climatic conditions [3,46,47]. Dark-colored honey has a higher mineral content than light-colored [48]. Some previous studies have shown that the mineral composition of honey correlates with the honey color and electrical conductivity [3,33].

Research conducted by [5] on seven types of honey from Serbia (linden, rapeseed, polyflora, honeydew, sunflower, phacelia and acacia) highlighted the highest total mineral content in linden honey, polyflora and honeydew. The same authors pointed out the predominant presence of potassium, magnesium and sodium (potassium 46.35–466.69 mg/kg, magnesium 5.71–72.31 mg/kg and sodium 6.75–160.04 mg/kg) in the analysed honey. Studies on polyfloral honey in Argentina, [49] confirm the presence of potassium as a predominant macroelement 90.92–1955.75 mg/kg, followed by calcium 18.60–136.14 mg/kg, sodium 6.10–89.98 mg/kg and magnesium 6.01–46.57 mg/kg. Samples of honeydew from Poland, analysed by [50], recorded average values of potassium from 2088 mg/kg to 2950 mg/kg, of calcium of 11.21–115.78 mg/kg, of magnesium of 33.53–65.10 mg/kg and sodium of 8.19–16.44 mg/kg.

In the honey samples analysed by us, the largest share of macroelements was occupied by potassium with 56.74–85.70 mg/kg, followed by calcium 32.52–70.54 mg/kg, magnesium 34.96–40.70 mg/kg and sodium 5.86–13.02 mg/kg (Table 1), and these results can be included in the data found in the literature. The highest content of macroelements was identified in linden honey and honeydew.

The microelements Fe, Cu, Zn and Mn have an important role in the physiological processes of the body. The honey analysed by us had values of iron in the range of 4.085 mg/kg for plain polyfloral honey up to 8.457 mg/kg (linden honey) and copper between 3.947 mg/kg for plain polyfloral honey and 6.986 mg/kg for acacia honey. The authors of [5] reported values of iron ranging from 0.79 to 5.51 mg/kg, of copper from 0.53 mg/kg to 1.6 mg/kg, of manganese between 0.21 mg/kg and 7.97 mg/kg and in the case of zinc, 0.38–20.36 mg/kg. The authors of [51] reported higher iron values for the four types of honey analysed (acacia, polyflora, linden and sunflower) from 19.38 mg/kg to 28.28 mg/kg and lower in the case of copper (0.18–0.33 mg/kg). The authors of [50] reported higher manganese content for honeydew in Poland (5.18–9.94 mg/kg).

In the honey samples analysed by us, the zinc and manganese content had the values of 2.78–4.55 mg/kg and 0.55–4.99 mg/kg, respectively, and were similar to those presented in the literature; the copper ones were slightly higher compared to those from the literature. High zinc and copper content was identified in acacia honey and manganese in honeydew (Table 1).

The collection of heavy metals together with nectar makes bees important indicators of environmental pollution [52]. Crowded car traffic but also other categories of pollutants can cause pollution of flower nectar and, implicitly, of honey [53]. Heavy metals (nickel, cadmium, lead) entering the food chain are toxic and can cause intoxication, allergies, chromosome changes and tumors [54].

In the honey we analysed, nickel was present in the range of 0.12–0.24 mg/kg, similar to the values recorded by [55]. The chromium content of the samples we analysed was between 0.10 mg/kg and 0.11 mg/kg, similar to the data reported by [53]. In the case of lead, the honey analysed by us had a content within the limits of 0.07–0.16 mg/kg, the higher content being recorded in the case of knotweed honey (Table 1). Similar data on lead content were reported by [56] for the honey from Cluj County, Romania. High values of lead content, from 0.76 mg/kg to 3.41 mg/kg, were reported by [57] for polyfloral honey from the region of Copșa Mică, Romania. For polyfloral honey from Italy, [58] reported a lead content of 1.10–1.74 mg/kg.

The cadmium content of the honey samples studied by us was in line with European legislation [59] with the exception of knotweed honey 0.13 mg/kg (Table 1), in which we found a slight exceeding of the standard. Sources of cadmium can be represented by some fertilizers, mining and sewage sludge [51]. Research conducted by [50] showed a close correlation between soil acidity and cadmium absorption in the plant and nectar, but there are other factors that influence the solubility of mineral elements such as the content of minerals and organic matter in the soil, soil pH, soil temperature and humidity, and soil mechanical and physical properties [60,61].

Honey contains various acids that are responsible for its acidity and pH, providing protection against microbial contamination [62]. According to the EU Council Directive (2001), honey acidity must be below 50 meq/kg. The high level of acidity may indicate honey fermentation [3].

In this study, the acidity of the analysed samples ranged from 2.2 mg/100 g (acacia honey) to 5.8 mg/100 g (honeydew) (Figure 2). Similar results on the acidity of floral honey have been obtained by other authors [5,62,63]. The presence of different organic acids, geographical origin and harvest season can influence the acidity of honey [26,64]. The pH values of the analysed samples were between 3.3 (acacia honey) and 3.73 (sunflower and wild cherry honey) (Figure 3) with an average of 3.53. The average pH values of honey samples from Vojvodina (Serbia) ranged from 3.88 (sunflower honey) to 3.99 (acacia honey) [39] and those from Romania were 4.09 for polyfloral honey, 4.22 for rapeseed honey and 3.94 for sunflower honey [65].

Honey is rich in sugars, monosaccharides occupy about 75% and disaccharides 10–15% [3]. Monosaccharides are mainly fructose and glucose [66,67]. Sugars are responsible for its energy value, viscosity, granulation and hygroscopicity [68].

According to [35], the minimum amount of reducing sugar for floral honey is 60 g 100 g^−1^ and for honeydew it should not be less than 45 g 100 g^−1^. In studies conducted by [69], the reducing sugar content was in the range 61.1% and 79%. In the case of the samples analysed by us, the amount of reducing sugar (Figure 4) was within the standard, with values from 60.78% (wild cherry honey) and 72.56% (hill polyflora honey).

Honey flavonoids come from nectar, pollen and propolis. It is known that flavonoids together with other phenolic components intervene in the protection against free radicals, reduce the level of H_2_O_2_ and NO, showing antiinflammatory and antioxidant effects [70]. The flavonoid content of honey varies depending on the botanical origin and the year of harvest [65].

The flavonoid content of the samples we analysed ranged from 9.29 mg QE/100 g (wild cherry honey) to 263.86 mg QE/100 g (linden honey) (Table 2). In the case of knotweed honey, the flavonoid content was 24.32 mg QE/100 g, within the range of 20.0–55.0 mg QE/100 g obtained by [71]. Rapeseed honey recorded a flavonoid content of 29.95 mg QE/100 g, sunflower honey 269.03 mg QE /100 g and polyfloral honey 29.01–29.48 mg QE /100 g, being close to that recorded by other authors [62,65].

Linden honey had the highest content of flavonoids, 263.86 mg QE/100 g, among the samples analysed, these being above the values recorded in the literature [66]. Acacia honey had a flavonoid content of 92.42 mg QE/100 g, higher than that of other authors [66,72]. The flavonoid content of knotweed honey obtained by us was 29.95 mg QE/100 g, lower than that obtained by [62], but it belongs within the limits of 11.97–44.54 QE/100 g obtained by [72].

In the honey analysed by us, the total phenol content (TPC) was from 177.6 mgGAE /100 g (acacia honey) to 1159.3 mgGAE/100 g (knotweed honey) (Table 2), being similar to the results obtained by [73]. The TPC values of rapeseed honey, sunflower and polyflora were higher compared to the results obtained by other authors.

Studies conducted by [65] showed a total polyphenol content of 19.9 mg GAE/100 g^−1^ for rapeseed honey, 21.1 mg GAE/100 g^−1^ for sunflower honey and 20.3 mg GAE/100 g^−1^ for polyfloral honey. The total phenol content obtained by [8], in rapeseed honey and polyflora was from 170–330 mg GAE/100 g.

The knotweed honey we analysed had a total polyphenol content of 187 mg GAE/100 g, within the limits of 100−195 mg GAE/100 g obtained by [71]. The honeydew analysed by us had a total polyphenol content of 1159.3 mg GAE/100 g and this corresponds to the data obtained by other authors [73]. The wild cherry honey analysed by us recorded a total polyphenol content of 647.5 mg GAE/100 g, and we have not found studies on this type of honey in the literature.

### 2.2. Antioxidant Activity

Studies have shown that there is a close correlation between antioxidant capacity and the content of flavonoids and polyphenols that depend on the botanical origin [62,65,66,73,74]. Studies undertaken by [62] on sunflower, rapeseed, polyflora honey and honeydew showed values of DPPH between 16.03% (rapeseed honey) and 72.03% (honeydew). The authors of [62] analysed the antioxidant activity of rapeseed, sunflower and polyfloral honey obtaining values of DPPH between 55.4% (rapeseed honey) and 70.7% (polyfloral honey).

The DPPH values that have been reported for honeydew vary with the botanical origin and geographical area. Spanish honeydew recorded DPPH values of 52.9–95.6% [75], the one from Greece 56.8–72.4% [76], the one from Croatia 12.2–48.89% [77] and that from Serbia 75.89–79.1% [5].

In the honey samples analysed by us, the DPPH values were between 41.88% (acacia honey) and 79.20% (honeydew) (Table 2).

In the case of the sunflower honey we analysed, the DPPH values were higher compared to those obtained by [62] but close to the value obtained by [78]. High DPPH values were recorded for dark-colored honey, of 79.21% (honeydew) and 79.90% (linden honey). In the case of light-colored honey, the DPPH values recorded by us were 41.89% (acacia honey), 50.41% (wild cherry honey) and 59.81% (rapeseed honey).

### 2.3. Antimicrobial Activity

To interpret the results of the antimicrobial testing, we calculated two indicators: BGR/MGR and BIR/MIR, using the Equations (1 and 3) and (2 and 4) presented in 3.11.

Figure 5, Figure 6, Figure 7, Figure 8, Figure 9, Figure 10, Figure 11, Figure 12, Figure 13 and Figure 14 show the bacterial inhibition rate (BIR%)/mycelial inhibition rate (MIR%), calculated according to Equations (1 and 3), while Appendix A presents the MIC through the optical density (OD) reading of honey samples tested on the ATTC strains.

A table containing the BGR%/MGR% values (2), when different concentrations of honey samples were applied to the screened strains, is presented in the Appendix A.

Subsequently, the data presented are discussed on each type of honey individually. Figure 5 presents the results obtained for the knotweed honey sample. All the tested strains presented only a positive strain boosting effect; all the inhibitory results correlated with the increase in concentration. Negative inhibitory results were obtained on *S. flexneri* and *P. aeruginosa*, while a medium effect was recorded against *S. pyogenes*, *S aureus*, *S. typhimurium*, *H. influenzae* and *C. parapsilopsis*. The best results were obtained against *E.coli* (26.93%), *C. albicans* (44.61%), *L. monocitogenes* (32.93%) and *B. cereus* (21.28%).

Figure 6 presents the research values expressed as BIR/MIR% for linden honey. Different from acacia honey is the potentiating effect present on *S. flexneri* and *C. albicans* of linden honey. In this case, the effect is a negative strain-boosting effect; therefore, the effect decreases with increased concentration. If, in the case of *S. flexneri*, linden honey 10% proved BIR at −60.37%, at 25% the value obtained was −179.66%, presenting a strain mass growth stimulated by linden honey. A similar effect is present in the case of *C. albicans* but with lower values. At 10%, MIR was 30.25%, and at 25%, 4.64%. Even if the results obtained were positive, the trend presented was a negative one; the smaller concentration determines a better effect, which implies the synergistic effect of linden honey on the fungal strain.

Research conducted by [79] showed reduced antimicrobial activity (expressed in BIR/MIR) of linden honey against the bacteria *S. pyogenes*, *S. aureus*, *P. aeruginosa*, *S. typhimurium* and *C. parapsilopis*, and good antimicrobial activity against *S. flexneri* (8.53–17.66%), *E. coli* (5.63–15.65%) and *E. influenzae* (15.09–26.17%). The authors of [80] reported linden honey average MIC values of 7.3% against *S. aureus* and 11.5% against *P. aeruginosa*, and [81] observed high antimicrobial activity against *S. pneumoniae* (MIC 21.3–42.5%).

A positive strain-boosting effect but with negative values was recorded in the case of *S. aureus*, *P. aeruginosa*, *S. typhimurium*, *H. influenzae* and *C. parapsilopsis*. The inhibition value increased with the increase in concentration, but it did not reach the MIC. Positive inhibitory results were recorded for *S. pyogenes* at 25% (44.88%), *E. coli* (32.19%), *L. monocitogenes* (21.69%) and *B. cereus* (19.43).

Wild cherry honey proved a negative strain-boosting effect influenced by increased concentration regarding *S. flexneri*, *P. aeruginosa* and *L. Monocitogenes* (Figure 7). The inhibition rate (BIR/MIR%) was positive in the case of *E. coli*, *C. albicans* and *B. cereus*, with values going as high as 47.35%. Concerning *S. pyogenes*, *S. aureus*, *S. typhimurium* and *C. parapsilopsis*, the results obtained were positive but with lower values ranging from −7.61 to 11.49. In the literature, we did not find studies on the chemical composition or antimicrobial and antioxidant activity of wild cherry honey.

Figure 8 presents the graphical form of the results obtained for acacia honey on each ATCC strain tested. The most resistant strains were *S. aureus*, *S. flexneri* and *P. aeruginosa.* For each of the concentrations tested, the inhibitory results were negative, implying the synergistic effect of acacia honey with the bacteria. The evolution is in line with the concentration, presenting a positive strain-boosting effect, which means that the inhibition correlates with the concentration, but the concentration tested was not enough to determine the MIC. Concerning *S. pyogenes*, *S. typhimurium* and *H. influenzae*, the evolution is similar, starting with negative inhibitory values ranging from −100.20% up to −42.44% in acacia 10% but with a better inhibitory effect once the concentration is increased. Therefore, in the case of acacia 25%, the BIC proved to be 9.48% in *S. pyogenes*, 10.15% for *S. typhimurium* and 3.69% inhibition of *H. influenzae*.

Similar values were recorded for acacia honey on *E. coli*, *C. parapsilopsis* and *B. cereus*. The values obtained for 10% ranged from −3.32% to −0.25, while for the 25% concentration, the results were 3.67% for *E. coli*, 14.12% for *C. parapsilopsis* and 19.36% for *B. cereus*. Concerning the best results obtained in the case of acacia honey, the most sensitive ATCC strains proved to be *C. albicans* and *L. monocitogenes*, with BIR values at 25% concentration of 33.28% and 29.08%, respectively.

Similar results on the antimicrobial activity of acacia honey have been obtained by other authors [79,82]. Reduced antimicrobial activity of acacia honey against *S. aureus* and *P. aeruginosa* was observed by [5,83], who reported good antimicrobial activity of it against *E. coli* and a weaker activity against *C. albicans*.

Comparing the BIR percentages obtained for honeydew, the most sensitive ATCC strains were: *E. coli*, *C. albicans* and *B. cereus* (Figure 9). For *S. pyogenes*, *S. aureus*, *S. flexneri* and *P. aeruginosa*, the bacterial inhibition rate (BIR%), depending on the concentration tested (10%; 15%, 20% and 25%), was influenced by the concentration. Except for 25% on *S. pyogenes* (10.99%), the results obtained proved only negative values but with a positive inhibitory trend. In the case of *S. typhimurium* and *L. Monocitogenes*, honeydew determined a potentiating effect, the result being a strain mass growth correlated to the increase in concentration.

Other authors have reported good antimicrobial activity of honeydew against *S. aureus* [5,28,84], who observed high antimicrobial activity of honeydew against *S. aureus*, *S. epidermidis*, *E. coli*, *P. aeruginosa* and *P. mirabilis*.

Concerning the four oilseed rape honey concentrations tested (10%, 15%, 20% and 25%), the results presented in Figure 10 show the best antimicrobial effect recorded against *E. coli*, *S. typhimurium*, and *B. cereus* with values reaching 24.05%. The effect was contrary to the concentration increase, with values ranging from −76.9% to −137.42% for *S. pyogenes*, −40.41% to −121.45% for *P. aeruginosa*, 37.58% to −11.2% for *C. albicans* and 14.11% down to −18.63% for *B. cereus*. *S. aureus* and *S. flexneri* proved a positive strain-boosting effect, with negative values that decreased alongside the concentration increase, showing small inhibitory activity due to insufficient concentration.

Our results are similar in the case of rapeseed honey with those obtained by [79] in terms of antimicrobial activity against *S. typhimurium*, *E. coli* and *C. albicans*. Contrary to our findings, [79] reported good antimicrobial activity for *S. pyogenes*, *S. flexneri*, *P. aeruginosa* and *E. influenzae*. Differences between the results can be attributed to the source of honey (beekeepers, supermarket, organic stores) and its processing [85], being valid for all types of honey analysed.

Figure 11 summarizes the data regarding the sunflower honey antimicrobial activity. Sunflower honey was most effective against *E. coli*, *C. parapsilopsis*, *C. albicans*, *L. monocitogenes* and *B. cereus* with BIR values between 19.61% and 33.65%, obtained at 25% concentration tested. There was a positive strain-boosting effect on *S. pyogenes*, *S. aureus* and *P. aeruginosa* with negative values ranging from −147.86% to −11.9% for sunflower honey 10%, values which increased alongside the increase in concentration, achieving −30.46% at M3 25%. While for *S. flexneri* the inhibitory activity decreases alongside concentration, proving a negative strain boosting effect, with values of −57.22% for sunflower honey at 10%, at 25%, BIR was −129.66%. *S. typhimurium* and *H. influenzae* showed similar effects with values ranging from −26.63% to 10% and reached 65% at 25%.

The antimicrobial effect of sunflower honey at concentrations of 40–100% was highlighted by [86] on the bacteria *S. aureus ATCC 29213*, *E. coli ATCC 25922*, *S. enterica ATCC 10708*, *Y. enterocolitica ATCC 23715* and *B. subtilis ATCC 23857*. The authors of [87] reported good antimicrobial activity against *E. coli*, *B. subtilis*, *Micrococcus luteus* and *Proteus myxofaciens* for the sunflower honey at a concentration of 75%. Studies conducted by [88] highlighted the increased antimicrobial activity of sunflower honey against the bacteria *S. aureus*, *L. monocytogenes*, *Salmonella* and *E. coli.*

Phacelia honey proved to be one of the few honey samples that showed a negative boosting effect only in *L. monocitogenes* (−2.99%). Even if the results on *S. aureus*, *S. flexneri*, *P aeruginosa* and *H. influenzae* proved that the concentration tested was insufficient to determine MIC, the inhibitory trend showed that phacelia honey does inhibit the mass growth of the strain. Positive inhibitory results were obtained in the case of phacelia honey against *S. pyogenes* at 25%, *E. coli* and *C. parapsilopsis* at 20% and at 25%, *C. albicans* and *B. cereus* (Figure 12). The authors of [89] reported good antimicrobial activity of phacelia honey (*Phacelia tanacetifolia Benth*), originating in Poland, on *S. aureus* and *P. aeruginosa* comparable to manuka honey.

Hill polyfloral honey (Figure 13) showed a similar effect as wild cherry honey, the difference being made by the positive strain-boosting effect on *P. aeruginosa* with values ranging from −116.75% to −37.01% and the negative strain-boosting effect on *C. albicans*, with values starting at 34.62 and decreasing to 25.22. The best effect recorded for hill polyfloral honey was on *E. coli* (27.82%) and *B. cereus* (31.44%).

Data recorded for plain polyfloral honey showed a different pattern. Positive inhibitory results were obtained for *S. pyogenes* at 25%, *S. aureus* at 25%, and *E. coli*, *C. parapsilopsis*, *C. albicans* and B. cereus, with the highest value recorded as MIR% on *C. albicans* (49.05). A negative boosting effect was found against *S. flexneri* (values ranging from −61.33% to −70.60%) and against *S. typhimurium* (from −22.34% to −83.65%). The quantity of honey tested influenced the inhibitory values showing a boosting effect, with MIR decreasing with the increase in concentration (Figure 14).

Studies conducted by [90] on polyfloral honey and a mixture of polyfloral honey and buckwheat from Kazakhstan showed an antimicrobial effect against S. aureus and E. fecalis, and [82] reported the antimicrobial potential of polyfloral honey in the Transylvania region, Romania, on *S. aureus*, *P. aeruginosa* and *B. cereus*. In the case of this type of honey, the botanical and geographical origin, but also the conditions of harvesting, processing and storage, can influence its antimicrobial activity.

Summarising the data presented in the Figure 5, Figure 6, Figure 7, Figure 8, Figure 9, Figure 10, Figure 11, Figure 12, Figure 13 and Figure 14, *S. pyogenes* was influenced in almost all the samples tested but only at 25% concentration. *S. aureus* was inhibited with positive values only by sunflower honey 25%, plain polyfloral honey 25% and knotweed honey 25%. Of all the ATCC strains tested, *S. flexneri* and *P. aeruginosa* proved to be the most resistant; none of the honey samples tested achieved positive BIR/MIR% values. Furthermore, *S. typhimurium* and *H. influenzae* followed the gradient resistance. C. parapsilopsis was found to be medium affected, while *E.coli*, *C. albicans*, *L. monocitogenes* and *B. cereus* proved to be the most sensitive to the action of the ten honey samples tested.

## 3. Materials and Methods

### 3.1. Honey Samples

A total of 10 samples of different types of honey: knotweed (Fallopia japonica), linden (Tilia europea), wild cherry (Prunus avium subsp. Avium), acacia (Robinia pseudoacacia), honeydew, rapeseed (Brasica napus), sunflower (Helianthus annuus), phacelia (Phacelia), plain and hill polyflora, were harvested in 2021 from the region of Banat, Romania (Table 3). All samples were stored in glass jars at room temperature of 20 ± 5 °C. Analyses were performed at the Interdisciplinary Research Platform (PCI) belonging to the Banat’s University of Agricultural Sciences and Veterinary Medicine “King Michael I of Romania” from Timisoara.

### 3.2. Determination of Humidity

Determination of humidity was made by the oven drying method. From each type of honey, 5 g/samples were weighed and dried in the oven (BINDER GmbH, Tuttingen, Germany), at the temperature of 103 °C (SR 784–3/2009). After 24 h, the samples were removed from the oven, and after cooling they were weighed and the result was calculated according to the formula [91,92]
Humidity= [(G_1_ − G_2_)/ G_1_ − G_3_)] × 100 (%)
Dry matter = 100 − Humidity (%)
where:

G_1_—the weight of petri dish and sample before drying (g);

G_2_—the weight of petri dish and sample after drying (g);

G—the weight of Petri dish (g).

### 3.3. Determination of Impurities

Each sample of honey (10 g/sample) was dissolved in 50 mL of water and homogenized in a shaker (Holt plate, Freising, Germany) for 30 min. The honey solutions obtained were filtered through filter paper (previously weighed). The samples obtained were placed in the oven at 103 °C for 10 min to dry the filter paper, then weighed [92]. The impurities were calculated according to the formula
I = (m_1_/m_2_) × 100 (%)
where:

I—represents the quantity of impurities (%);

m_1_—represents the mass of the sample taken for analysis (g);

m_2_—represents the mass of residue left on the filter paper after drying (g).

### 3.4. Determination of Mineral Substance Content (ash)

The honey samples (3 g honey/sample) were placed in the calcination furnace (Nabertherm, Lilienthal, Germany) at the temperature of 525 °C to the constant mass. After cooling, the ash crucibles were weighed and, to calculate and express the results, we applied the formula [91,92]
Ash = (m − m_1_)/m_2_ − m_1_) (%)
where:

m—represents the mass of the melting pot with the ash obtained after calcination (g);

m_1_—represents the mass of the empty melting pot (g);

m_2_—represents the mass of the melting pot with honey (g).

After calcination, 10 mL of hydrochloric acid were added to each sample, after which the samples thus obtained were transferred to glass tubes in order to determine the content of micro and macro elements [93]. We prepared 10 graduated flasks of 50 mL, the samples were filtered and brought to the mark with water. A multielement standard solution was used for calibration, Centipur Merk. The determinations were performed on the atomic absorption spectrophotometer Varian AA 240FS, each element being read according to the Table 4.

### 3.5. Determination of Acidity

In order to determine the acidity, 50 mL of water and 2 drops of Phenolphthalein were added to each sample of honey (10 g/sample). The samples thus prepared were introduced into the stirrer Holt plate IDL, Freising, Germany, for 30 min. After dissolution, the samples were filtered through filter paper, then titrated with sodium hydroxide 0.1 n solution until the pink color persisted for 30 s [92,94]. To calculate and express the results, we used the formula
Acidity = [ (V × 0.1)/10] × 100 (ml NaOH 0.1 n/100 g honey)
where:

V—represents the volume of sodium hydroxide solution used in the titration (mL);

0.1—represents the normality of sodium hydroxide solution used for titration.

### 3.6. Determination of pH

To determine the pH, we used the pH meter inoLab pH 730 (Xylem Analytics, Weilheim, Germany). Three g of honey/sample was dissolved in 30 mL of water and mixed with the stirrer Holt plate Stirrer LM4-1002 for 30 min [92,94]. The room temperature at which the pH of the samples was determined was between 23 and 24 °C, and the working range of the pH was −2.000 ± 19.999, with accuracy of ± 0.05.

### 3.7. Determination of Reducing Sugar

To determine the reducing sugar, we used the method described by [92]. From each type of honey, 3 g/sample was weighed, over which we added water up to 200 mL and mixed very well. Twenty mL of the resulting solution was extracted into a glass container; we added water up to 100 mL and homogenized again. The resulting solution was the working solution. In a bowl we put 20 mL of copper sulphate solution, 20 mL of alkaline Seignette salt solution and 20 mL of water and mixed. The pot was placed on the electric hob, and at the time of boiling we added 20 mL of the working solution. Five min after the boiling started, the dish was removed from the stove and cooled by immersing it into water. After cooling, 25 mL of sodium chloride solution was added, the solution in the flask becoming clear with a greenish-blue appearance after stirring. When 2 g of baking soda was added, after the effervescence had stopped, a residue of baking soda remained in the solution in the vessel, the color becoming intense blue. The solution obtained was titrated with iodine solution, stirring constantly. At the beginning of the titration the color of the solution was milky white, becoming clear and green at the end of the titration. The identification of the excess iodine was made by adding 0.5 starch solution to the green solution, the color becoming dark blue. This was then titrated with sodium thiosulphate solution until the color of the solution reached light blue. The reducing sugar content expressed in invert sugar was calculated according to the formula [94]
Cinvert sugar = [(m × 10 × 5)/(m_1_ × 1000)] × 100 (%)
where:

m—represents the amount of invert sugar (mg);

m_1_—represents the amount of honey analyzed (g);

10—represents the ratio between the volume of the solution in the 200-mL volumetric container and the volume of the solution taken for dilution;

5—represents the ratio between the volume of the solution in the 100-mL volumetric container and the volume of the diluted solution taken for analysis.

### 3.8. Determination of Total Phenolic Content (TPC)

Of each type of honey, we weighed 1 g/sample in a container with a lid, over which we added 10 mL of alcohol 70%. The samples were homogenized with the stirrer Holt plate Stirrer (IDL, Freising, Germany) for 30 min, after which they were filtered with filter paper. In glass tubes we added 0.5 mL of the filtered sample, and titrated with 1.25 mL reagent Folin–Ciocalteu (Sigma-Aldrich Chemic GmbH, München, Germany), diluted 1:10 with distilled water. The samples thus prepared were incubated for 5 min at room temperature, we added 1 mL Na_2_CO_3_ (60 g/L aqueous solution) and were then introduced to the thermostatic incubator (Memmert GmbH, Schwabach, Germany) at 50 °C for 30 min. At the expiration of the time, the absorbance at 750 nm was read with a spectrometer UV–VIS (Analytical Jena Specord 205, Jena, Germany), using ethanol as control. The calibration curve was obtained using gallic acid (concentration range: 2.5–250 μg/mL. The results were expressed in mg GA per g dry matter (d.m.). All determinations were performed in triplicate [92,95].

### 3.9. Determination of Flavonoid Content (FC)

To determine the flavonoids, we prepared 10 containers with lids, in which we inserted 1 g of honey sample and 10 mL of 60% alcohol. They were homogenized with Holt plate Stirrer for 30 min, after which they were filtered with filter paper. In 10 colorless glass containers, we added 1.5 mL of the previously prepared extract, 4.5 mL H_2_O and 1 mL NaNO_2_, after which they were left to incubate for 6 min. After incubation we added 1 mL Al(NO_3_)_3_ 10%, and the samples were left to incubate again for 6 min. When the incubation time had elapsed, we added 10 mL NaOH 4% and supplemented with alcohol 70% up to 25 mL [95]. The samples were left to stand for 15 min, after which the absorbance was read at 510 nm using the spectrometer UV–VIS (Analytical Jena Specord 205, Jena, Germany). The quercetin solution was used as control. The results were expressed in mg QE/100 g and all determinations were performed in triplicate [92,96].

### 3.10. Determination of Antioxidant Activity (AA) by DPPH

The determination of the antioxidant activity of the 10 honey samples by the DPPH method was performed according to the method described by [95]. From each type of honey studied, we weighed 1 g of honey/sample, diluted with 10 mL of 60% alcohol, and then we filtered it through filter paper. The obtained extracts were left to incubate for 30 min. For the negative control, we used 60% ethanol; the sample was read on the spectrometer UV–VIS (Analytical Jena Specord 205, Jena, Germany) at the absorbance of 518 nm. For the positive control, we introduced into a test tube 1 mL of solution DPPH (3 mM), 2.5 mL extract, left it for incubation for 30 min and then read it on the spectrometer (absorbance 518). For the blank, we placed in a test tube 1 mL of ethanol and 2.5 mL of extract, left it to incubate for 30 min, then read it at the absorbance 518, using a spectrometer. The calculation of the antioxidant activity of honey was analysed with the formula
$$AA=100-\left\{\frac{\left[\left(ABS_{sample}-ABS_{blank}\right)\cdot 100\right]}{ABS_{control}}\right\}\;$$
where:

AA—represents the antioxidant activity of the analysed sample;

ABS_sample_—represents the absorbance of the sample measured at a wavelength of 518 nm;

ABS_control_—represents the absorbance of the DPPH sample measured at a wavelength of 518 nm;

ABS_blank_—represents the absorbance of the alcohol sample measured at a wavelength of 518 nm [97].

### 3.11. Antimicrobial Activity

Aqueous extracts of each honey sample were prepared by mixing 0.5 g of honey with 1 mL of sterile distilled water, and, subsequently, different quantities were spotted into 96-well plates to reach the 10%, 15%, 20% and 25% concentrations.

The microbial reference strains (ATCC) used in this study were obtained from the culture collection of the Microbiology Laboratory of the Interdisciplinary Research Platform within the University of Agricultural Sciences and Veterinary Medicine “King Mihai I of Romania” in Banat, Timisoara.

The honey samples were tested against the following reference strains: Streptococcus pyogenes (ATCC 19615), Staphylococcus aureus (ATCC 25923), Shigella flexneri (ATCC 12022), Pseudomonas aeruginosa (ATCC 27853), Escherichia coli (ATCC 25922), Salmonella typhimurium (ATCC 14028), Haemophilus influenzae type B (ATCC 10211), Candida albicans (ATCC 10231), Candida parapsilopsis (ATCC 22019), Listeria monocytogenes (ATCC 19114) and Bacillus cereus (ATCC 10876).

The MIC is defined as the lowest compound concentration that yields no visible microorganism growth. Our previous research described the method of MIC determination based on the microbial mass loss by measurement of OD by spectrophotometry according to ISO 20776-1:2019 [79,92].

#### 3.11.1. Bacterial culture

A 10^−3^ dilution of the fresh culture was used to perform the assay, an inoculum equivalent to a 0.5 McFarland standard. The bacterial strains were revived by overnight growth in brain heart Infusion (BHI) broth (Oxoid, CM1135) at 37 °C and, subsequently, passed on BHI Agar (Oxoid, CM1136) for 24 h at 37 °C. The cultures were then diluted at an optical density (OD) of 0.5 McFarland standard (1.5  ×  10^8^ UFC × mL) using BHI broth and evaluated with a McFarland Densitometer (Grand-Bio, England). The dilutions were spotted at 100 μL in each well of the 96 microdilutions well plate, using a Calibra digital 852 multichannel pipette. The tested honey samples were added into wells at 10%, 15%, 20% and 25%. The plates were covered and left 24 h at 37 °C. After 24 h, the OD was measured at 540 nm using an ELISA reader (BIORAD PR 1100, Hercules, CA, USA). Triplicate tests were performed for all samples. The suspensions of strain and BHI were used as a negative control.

For interpretation, two indicators were calculated, BGR and BIR, by using the Equations (1) and (2)
(1)$$\mathrm{BGR}=\frac{OD_{sample}}{OD_{negativecontrol}}\times 100(\%)$$

(2)
BIR= 100 − BGR (%)

where:

OD sample—optical density at 540 nm as the mean value of triplicate readings for the samples tested in the presence of the selected bacteria;

OD negative control—optical density at 540 nm as the mean value of triplicate readings for the selected bacteria in BHI.

#### 3.11.2. Fungal Culture

A 10^−2^ dilution of the fresh culture was used to perform the assay, an inoculum equivalent to a 0.5 McFarland standard. The ATCC fungal strains were revived by overnight growth in brain heart infusion (BHI) broth (Oxoid, CM1135) at 37 °C and, subsequently, passed on BHI Agar (Oxoid, CM1136) for 48 h at 37 °C. The cultures were then diluted at an OD of 0.5 McFarland standard using BHI broth, a value determined by using a McFarland Densitometer (Grand-Bio, England). The honey samples were tested by placing 100 μL of fungal suspension into each well of the 96 microdilution wells plate. The tested honey samples were added into wells at 10%, 15%, 20% and 25%. The plates were covered and left for 48 h at 37 °C. After 48 h, the OD was measured at 540 nm. Triplicate tests were performed for all samples.

For interpretation of the results, two indicators were calculated, MGR and MIR, using the following Equations (Equations (3) and (4))
(3)$$\mathrm{MGR}=\frac{OD_{sample}}{OD_{negativecontrol}}\times 100(\%)$$

(4)
MIR=100 − MGR (%)

where:

OD sample—optical density at 540 nm as the mean value of triplicate readings for the samples tested in the presence of the selected fungi;

OD negative control—optical density at 540 nm as the mean value of triplicate readings for the selected fungi in BHI.

### 3.12. Statistical Analysis

The results presented are the average of the values obtained with the standard deviation (SD). All calculations were obtained using the statistical program IBM SPSS 22. The statistical differences (*p* < 0.05) between the analysed honey types were processed using the ANOVA with Tukey′s test.

## 4. Conclusions

The chemical composition of the honey of knotweed (Fallopia japonica), linden (Tilia europea), wild cherry (Prunus avium subsp. Avium), acacia (Robinia pseudoacacia), honeydew, oilseed rape (Brasica napus), sunflower (Helianthus annuus), phacelia (Phacelia), plain polyflora and hill polyflora presents a great variability, being conditioned by the botanical origin. All the samples analysed had values of acidity and pH, but also of the content of impurities, within the limits allowed by the quality standards. The mineral content was higher for dark-colored honey (linden and honeydew) compared to light-colored honey (acacia and wild cherries).

The content of flavonoids and polyphenols is responsible for the antioxidant activity of honey, highlighting in the case of our study, the linden honey and honeydew, which recorded the best antioxidant activity.

The present study regarding the chemical attributes and the antimicrobial potential of honey confirms previous researches on the impact of botanical and geographical origin, bee genetics and meteorological factors, but also to the conditions of harvesting, processing and storage on the honey quality.

## Figures and Tables

**Figure 1 molecules-27-04179-f001:**
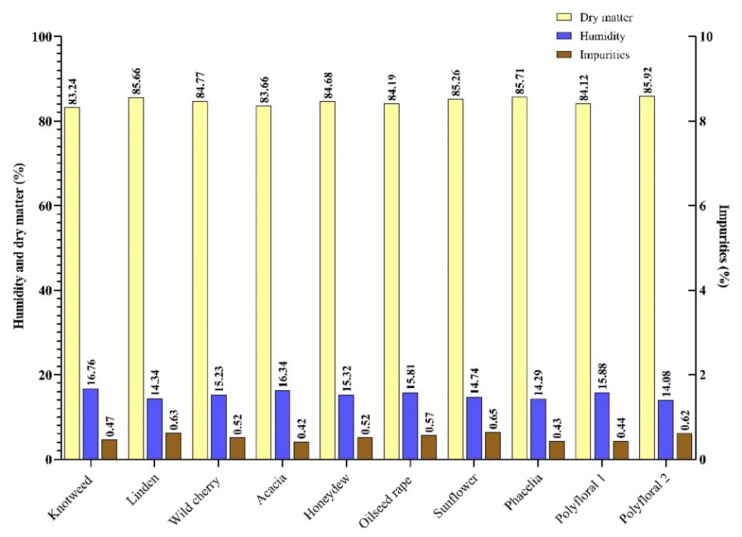
Humidity (H), dry matter (D.M.) and impurities (I) content (%) of the honey samples.

**Figure 2 molecules-27-04179-f002:**
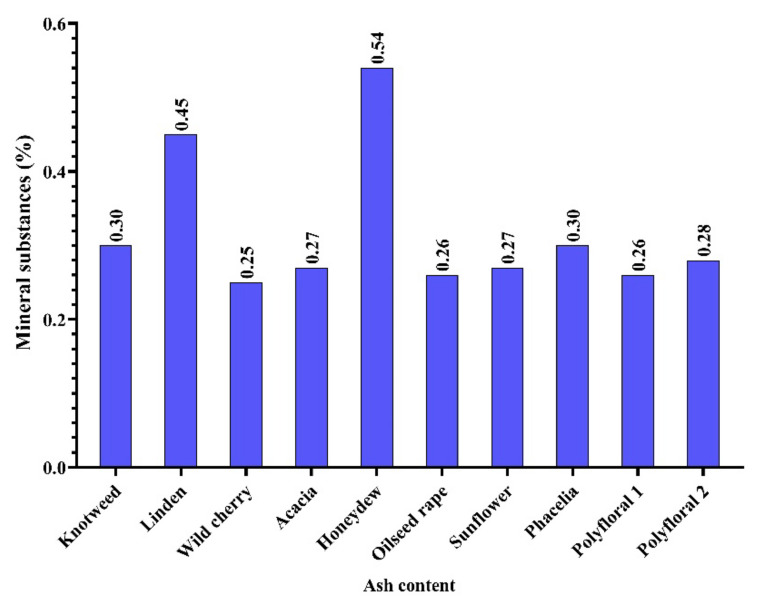
Mineral substance (ash) content (%).

**Figure 3 molecules-27-04179-f003:**
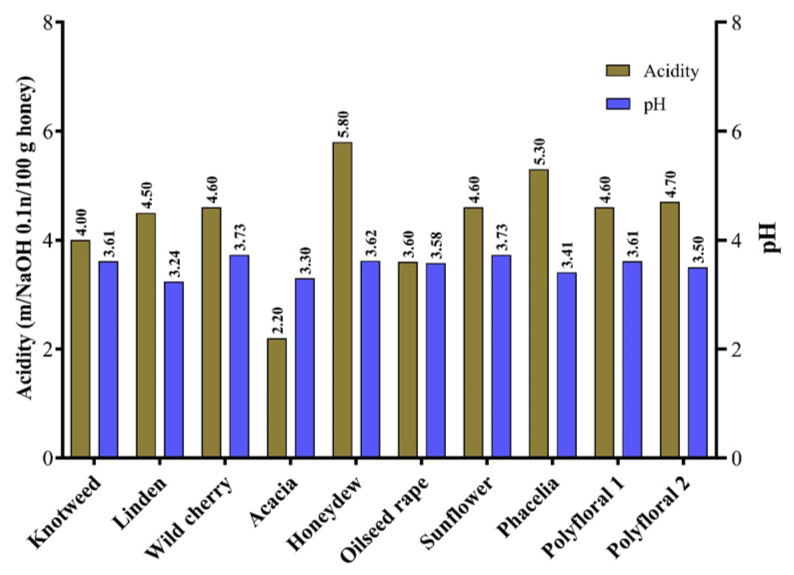
The acidity and pH of the honey samples.

**Figure 4 molecules-27-04179-f004:**
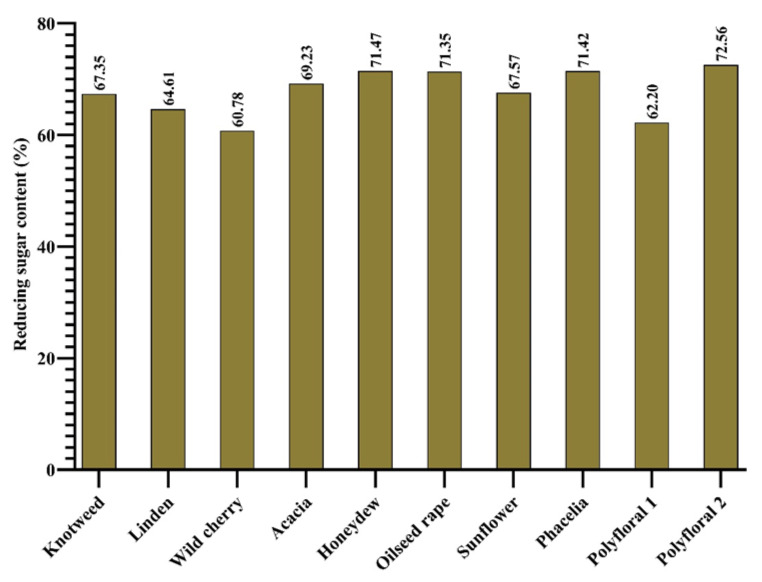
Reducing sugar content (%).

**Figure 5 molecules-27-04179-f005:**
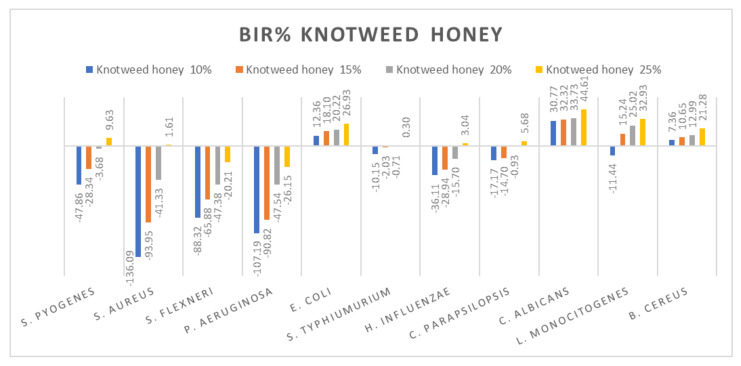
Knotweed honey antimicrobial activity (expressed as BIR%/MIR%) on ATCC.

**Figure 6 molecules-27-04179-f006:**
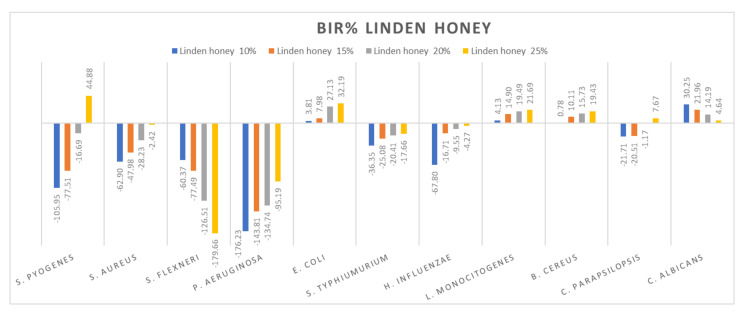
Linden honey antimicrobial activity (expressed as BIR%/MIR%) on ATCC strains.

**Figure 7 molecules-27-04179-f007:**
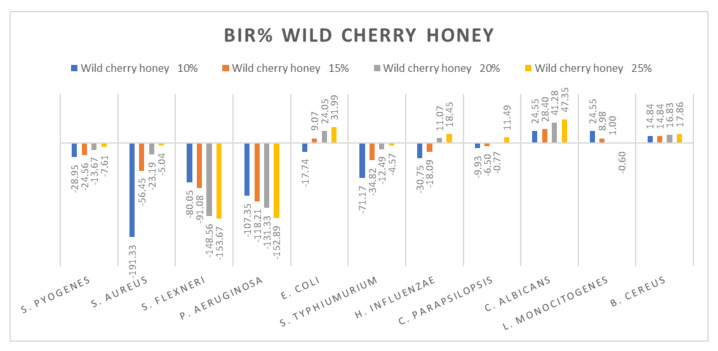
Wild cherry antimicrobial activity (expressed as BIR%/MIR%) on ATCC strains.

**Figure 8 molecules-27-04179-f008:**
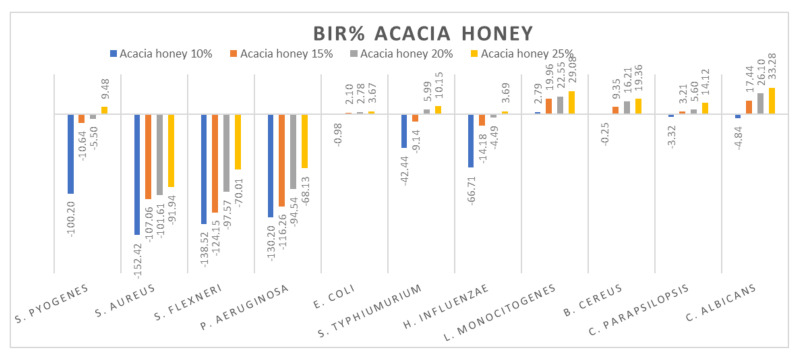
Acacia honey antimicrobial activity (expressed as BIR%/MIR%) on ATCC strains.

**Figure 9 molecules-27-04179-f009:**
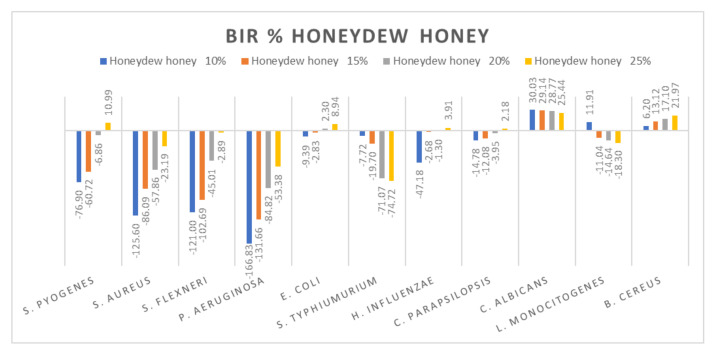
Honeydew antimicrobial activity (expressed as BIR%/MIR%) on ATCC strains.

**Figure 10 molecules-27-04179-f010:**
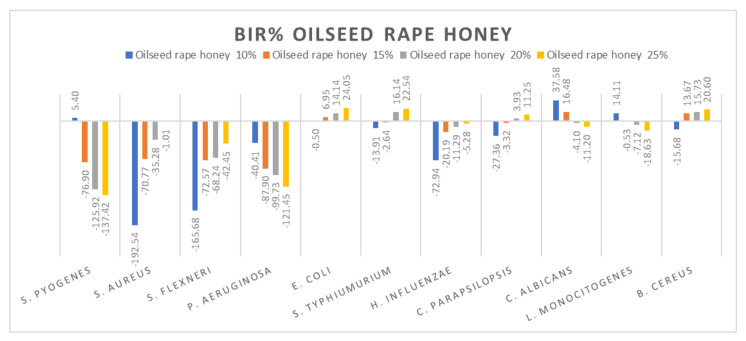
Oilseed rape honey antimicrobial activity (expressed as BIR%/MIR%) on ATCC strains.

**Figure 11 molecules-27-04179-f011:**
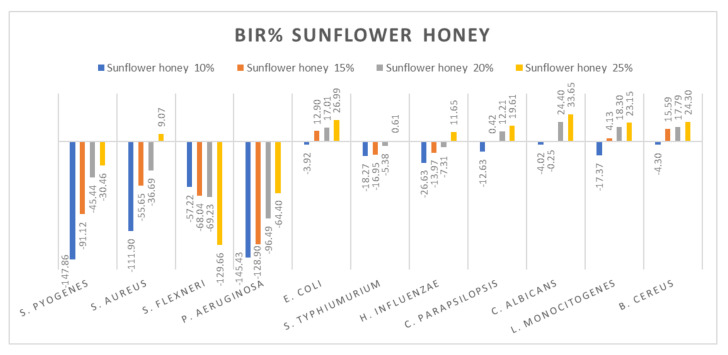
Sunflower honey antimicrobial activity (expressed as BIR%/MIR%) on ATCC strains.

**Figure 12 molecules-27-04179-f012:**
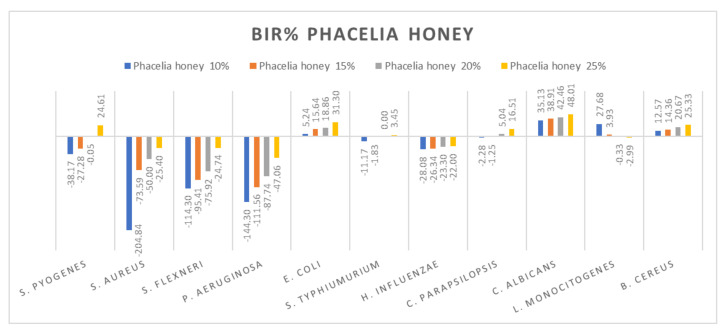
Phacelia honey antimicrobial activity (expressed as BIR%/MIR%) on ATCC.

**Figure 13 molecules-27-04179-f013:**
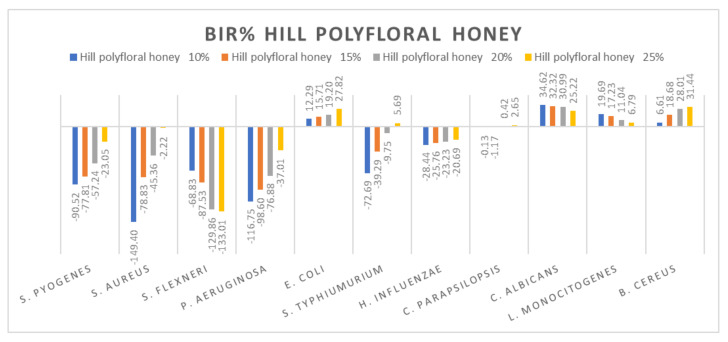
Hill polyfloral honey antimicrobial activity (expressed as BIR%/MIR%) on ATCC strains.

**Figure 14 molecules-27-04179-f014:**
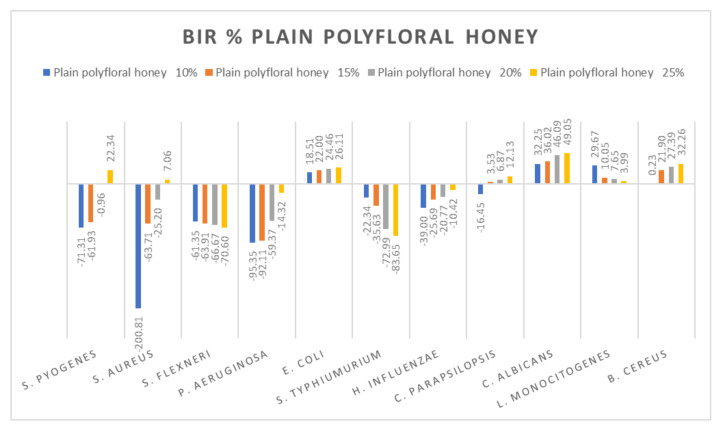
Plain polyfloral honey antimicrobial activity (expressed as BIR%/MIR%) on ATCC.

**Table 1 molecules-27-04179-t001:** The macro and micro-mineral content of the honey in the region of Banat, Romania (mg/kg).

Type of Honey	K	Ca	Mg	Na	Fe	Zn	Mn	Cu	Ni	Pb	Cr	Cd
	$\bar{x}\pm SD$	$\bar{x}\pm SD$	$\bar{x}\pm SD$	$\bar{x}\pm SD$	$\bar{x}\pm SD$	$\bar{x}\pm SD$	$\bar{x}\pm SD$	$\bar{x}\pm SD$	$\bar{x}\pm SD$	$\bar{x}\pm SD$	$\bar{x}\pm SD$	$\bar{x}\pm SD$
Knotweed	81.332 ^j^ ±0.002	32.521 ^j^ ±0.0002	35.280 ^j^ ±0.0001	7.673 ^j^ ±0.0001	4.261 ^a^ ±0.0002	3.133 ^i^ ±0.0058	0.954 ^h^ ±0.003	4.272 ^j^ ±0.001	0.220 ^j^ ±0.001	0.163 ^g^ ±0.010	0.114 ^a^ ±0.001	0.130 ^j^ ±0.001
Linden	85.706 ^b^ ±0.002	70.547 ^b^ ±0.0002	40.700 ^b^ ±0.0001	12.510 ^b^ ±0.000	8.457 ^b^ ±0.0002	3.881 ^b^ ±0.0001	1.345 ^b^ ±0.003	5.139 ^b^ ±0.001	0.233 ^b^ ±0.001	0.076 ^b^ ±0.010	0.116 ^b^ ±0.001	0.049 ^b^ ±0.001
Wild cherry	74.364 ^f^ ±0.002	39.549 ^f^ ±0.0002	33.883 ^f^ ±0.0001	6.512 ^f^ ±0.0001	5.938 ^c^ ±0.0002	3.255 ^f^ ±0.0001	1.125 ^f^ ±0.003	3.902 ^f^ ±0.001	0.155 ^f^ ±0.001	0.111 ^a^ ±0.010	0.105 ^f^ ±0.001	0.089 ^f^ ±0.001
Acacia	56.749 ^a^ ±0.002	37.370 ^a^ ±0.0002	35.179 ^a^ ±0.0001	13.025 ^a^ ±0.0001	7.284 ^d^ ±0.0002	4.550 ^a^ ±0.0001	0.902 ^a^ ±0.003	6.986 ^a^ ±0.001	0.249 ^a^ ±0.001	0.109 ^a^ ±0.010	0.114 ^a^ ±0.001	0.078 ^a^ ±0.001
Honeydew	82.367 ^e^ ±0.002	67.473 ^e^ ±0.0002	39.846 ^e^ ±0.0001	7.591 ^e^ ±0.0001	6.237 ^e^ ±0.0002	2.780 ^e^ ±0.0001	4.999 ^e^ ±0.003	4.505 ^e^ ±0.001	0.199 ^e^ ±0.001	0.118 ^d^ ±0.010	0.110 ^e^ ±0.001	0.099 ^e^ ±0.001
Oilseed rape	78.076 ^d^ ±0.002	41.440 ^d^ ±0.0002	35.387 ^d^ ±0.0001	4.671 ^d^ ±0.0001	3.934 ^f^ ±0.0002	3.121 ^d^ ±0.0001	0.720 ^d^ ±0.003	4.000 ^d^ ±0.001	0.209 ^d^ ±0.001	0.118 ^d^ ±0.010	0.108 ^d^ ±0.001	0.028 ^d^ ±0.001
Sunflower	65.089 ^c^ ±0.002	54.280 ^c^ ±0.0002	38.097 ^c^ ±0.0001	8.203 ^c^ ±0.0001	7.218 ^g^ ±0.0002	3.177 ^c^ ±0.0001	0.551 ^c^ ±0.003	5.037 ^c^ ±0.001	0.202 ^c^ ±0.000	0.131 ^c^ ±0.010	0.108 ^c^ ±0.001	0.061 ^c^ ±0.001
Phacelia	73.078 ^i^ ±0.002	42.825 ^i^ ±0.0002	38.865 ^i^ ±0.0001	6.535 ^i^ ±0.0001	4.903 ^h^ ±0.0002	3.074 ^h^ ±0.0001	0.552 ^c^ ±0.003	3.994 ^i^ ±0.001	0.163 ^i^ ±0.001	0.147 ^e^ ±0.010	0.118 ^i^ ±0.001	0.024 ^i^ ±0.001
Polyfloral 1	76.917 ^g^ ±0.002	40.490 ^g^ ±0.0002	34.961 ^g^ ±0.0001	5.865 ^g^ ±0.0001	4.592 ^i^ ±0.0002	4.356 ^g^ ±0.0001	0.769 ^g^ ±0.003	5.056 ^g^ ±0.001	0.171 ^g^ ±0.001	0.149 ^e^ ±0.010	0.106 ^g^ ±0.001	0.068 ^g^ ±0.001
Polyfloral 2	64.977 ^h^ ±0.002	44.503 ^h^ ±0.0002	36.409 ^h^ ±0.0001	6.206 ^h^ ±0.0001	4.085 ^j^ ±0.0002	2.783 ^e^ ±0.0001	0.551 ^c^ ±0.003	3.947 ^h^ ±0.001	0.129 ^h^ ±0.001	0.097 ^f^ ±0.010	0.107 ^h^ ±0.001	0.108 ^h^ ±0.001
P (Anova test)	0.000	0.000	0.000	0.000	0.000	0.000	0.000	0.000	0.000	0.000	0.000	0.000

All results are expressed as means of triplicate ± standard deviation (SD). Between means with the same letter *p* > 0.05; between means with different letters *p* < 0.05.

**Table 2 molecules-27-04179-t002:** The content of polyphenols, flavonoids and the antioxidant capacity of the honey from the region of Banat, Romania.

Type of Honey	Polyphenols $\bar{x}\pm SD$	Flavonoids $\bar{x}\pm SD$	Antioxidant Capacity (DPPH) $\bar{x}\pm SD$
Knotweed	187.00 ± 0.200 ^a^	24.32 ± 0.814 ^a, f^	57.22 ± 0.005 ^a^
Linden	781.10 ± 0.200 ^b^	263.86 ± 0.814 ^b, c^	79.89 ± 1.853 ^b^
Wildcherry	647.50 ± 0.200 ^c^	9.29 ± 0.030 ^d^	50.40 ± 0.005 ^c^
Acacia	177.60 ± 0.200 ^d^	92.42 ± 0.010 ^e^	41.88 ± 0.025 ^d^
Honeydew	1159.30 ± 0.300 ^e^	29.02 ± 0.020 ^a, f^	79.20 ± 0.066 ^b^
Oilseed rape	496.80 ± 0.100 ^f^	29.95 ± 0.808 ^a, f^	59.80 ± 0.110 ^e, f^
Sunflower	854.10 ± 0.400 ^g^	269.43 ± 0.760 ^a, c^	77.07 ± 0.037 ^g^
Phacelia	910.00 ± 0.300 ^h^	26.20 ± 0.030 ^f^	66.71 ± 0.390 ^h^
Hill polyfloral honey	350.80 ± 0.300 ^i^	29.48 ± 0.808 ^f^	58.56 ± 0.020 ^a, f^
Plain polyfloral honey	565.90 ± 0.200 ^j^	29.01 ± 6.455 ^f^	57.25 ± 0.040 ^a^
Mean	613.01	80.30	62.80
SD	321.41	100.60	12.74
Min	177.60	9.29	41.88
Max	1159.30	269.43	79.89
P (Anova test) *	0.000	0.000	0.000

* All results are expressed as means of triplicate ± standard deviation (SD). Between means with the same letter *p* > 0.05; between means with different letters *p* < 0.05.

**Table 3 molecules-27-04179-t003:** The botanical and geographical origin of the honey samples from the region of Banat, Romania.

No. Sample	Botanical Origin	Geographical Origin
1	Knotweed (*Fallopia japonica*)	Caransebeș
2	Linden (*Tilia europea*)	Timișoara
3	Wild cherry (*Prunus avium subsp. Avium)*	Radimna
4	Acacia (*Robinia pseudoacacia*)	Caransebeș
5	Honeydew	Reșița
6	Oilseed rape (*Brasica napus*)	Sânandrei
7	Sunflower (*Helianthus annuus*)	Sânandrei
8	Phacelia (*Phacelia*)	Sânandrei
9	Plain polyfloral	Sânandrei
10	Hill polyfloral	Caransebeș

**Table 4 molecules-27-04179-t004:** The parameters used to read the mineral elements.

Symbol	Line ʎ (nm)	Lanpenstrom (mA)	Spalt (nm)
Cu	324.8	4	0.5
Ca	422.7	10	0.5
Ni	232.0	4	0.2
Fe	248.3	5	0.2
Pb	217.0	10	1.0
Na	589.0	3	0.8
Cr	357.9	8	0.2
Zn	213.9	5	1.0
K	766.5	4	0.8
Mn	279.5	5	0.2
Cd	228.8	4	0.5
Mg	285.2	4	0.5

## Data Availability

The report of the analyses performed for the samples in the paper can be found at the Interdisciplinary Research Platform (PCI) belonging to the Banat University of Agricultural Sciences and Veterinary Medicine “King Michael I of Romania” in Timisoara.

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
