# Peer review of "Chemical Composition, Antioxidant and Antimicrobial Activity of Some Types of Honey from Banat Region, Romania"

_molecules, 2022, doi:10.3390/molecules27134179_

Round 1
Reviewer 1 Report
Review Report
Manuscript ID: molecules-1774084
Title: Chemical composition, antioxidant and antimicrobial activity of some honey sorts from Banat region, Romania
Journal: Molecules
In this extensive study, the chemical characterization of ten types of honey from the Banat region, Romania has been thoroughly conducted, along with the antioxidant activity and antimicrobial potential of these samples. Honey samples were tested on the water content, dry matter, impurities, acidity and pH, the content of reducing sugar, minerals, flavonoids, and the total phenolic content. Authors have shown that the analyzed honey samples possess good nutritional characteristics. From the analysis of the mineral content of the honey samples variability of macro and microminerals was observed and correlated by its botanical origin.
Reviewer’s suggestions:
Line 117-118 „Dark color honey has a higher mineral content than the light-colored one” is there some scientific explanation for this connection between color and mineral content.
Line 136-138: Instead of decimal commas authors have to use decimal points.
Line 253-254 and 263-265: The DPPH values' numbers do not match the one presented in Table 2, and always by 1 %. Is there some particular reason for this?
This study has the potential to be cited.
Author Response
Dear Reviewer,
We would like to address all our thanks and gratitude for the constructive observations, corrections and recommendations.

Reviewer 2 Report
Please see the attached document

Author Response

(The authors gave the same response as above.)

Round 2
Reviewer 2 Report
After the changes, the article is ready for publication.